# Plant Microbiome Engineering: Hopes or Hypes

**DOI:** 10.3390/biology11121782

**Published:** 2022-12-07

**Authors:** Muhammad Siddique Afridi, Sher Ali, Abdul Salam, Willian César Terra, Aqsa Hafeez, Baber Ali, Mona S. AlTami, Fuad Ameen, Sezai Ercisli, Romina Alina Marc, Flavio H. V. Medeiros, Rohini Karunakaran

**Affiliations:** 1Department of Plant Pathology, Federal University of Lavras, (UFLA), Lavras 37200-900, MG, Brazil; 2Department of Food Engineering, Faculty of Animal Science and Food Engineering, University of São Paulo (USP), Pirassununga 13635-900, SP, Brazil; 3Zhejiang Key Laboratory of Crop Germplasm, Department of Agronomy, College of Agriculture and Biotechnology, Zhejiang University, Hangzhou 310058, China; 4Department of Plant Sciences, Quaid-i-Azam University, Islamabad 45320, Pakistan; 5Department of Biotechnology, Quaid-i-Azam University, Islamabad 45320, Pakistan; 6Biology Department, College of Science, Qassim University, Burydah 52571, Saudi Arabia; 7Department of Botany and Microbiology, College of Science, King Saud University, Riyadh 11451, Saudi Arabia; 8Department of Horticulture, Faculty of Agriculture, Ataturk University, 25240 Erzurum, Turkey; 9Food Engineering Department, Faculty of Food Science and Technology, University of Agricultural Science and Veterinary Medicine Cluj-Napoca, 3-5 Calea Mănă ¸stur Street, 400372 Cluj-Napoca, Romania; 10Unit of Biochemistry, Centre of Excellence for Biomaterials Engineering, Faculty of Medicine, AIMST University, Semeling, Bedong 08100, Malaysia; 11Department of Computational Biology, Institute of Bioinformatics, Saveetha School of Engineering (SSE), SIMATS, Thandalam, Chennai 602105, Tamil Nadu, India; 12Centre of Excellence for Biomaterials Science, AIMST University, Semeling, Bedong 08100, Malaysia

**Keywords:** plant microbiome, complexity, biotic and abiotic hurdles, suppressive soil, microbiome engineering

## Abstract

**Simple Summary:**

Plant microbiome is a key determinant of plant health and productivity. Plant microbiome is an alternative untapped source that could be harnessed for plant health and productivity. Microbiome engineering aims to manipulate the microbiome toward a particular community that will enhance important plant functions. In this article, we review the plant microbiome composition, microbial diversity, complex plant microbiome interaction and major challenges that serve as bottlenecks and discourage the approaches of plant microbiome engineering.

**Abstract:**

Rhizosphere microbiome is a dynamic and complex zone of microbial communities. This complex plant-associated microbial community, usually regarded as the plant’s second genome, plays a crucial role in plant health. It is unquestioned that plant microbiome collectively contributes to plant growth and fitness. It also provides a safeguard from plant pathogens, and induces tolerance in the host against abiotic stressors. The revolution in omics, gene-editing and sequencing tools have somehow led to unravel the compositions and latent interactions between plants and microbes. Similarly, besides standard practices, many biotechnological, (bio)chemical and ecological methods have also been proposed. Such platforms have been solely dedicated to engineer the complex microbiome by untangling the potential barriers, and to achieve better agriculture output. Yet, several limitations, for example, the biological obstacles, abiotic constraints and molecular tools that capably impact plant microbiome engineering and functionality, remained unaddressed problems. In this review, we provide a holistic overview of plant microbiome composition, complexities, and major challenges in plant microbiome engineering. Then, we unearthed all inevitable abiotic factors that serve as bottlenecks by discouraging plant microbiome engineering and functionality. Lastly, by exploring the inherent role of micro/macrofauna, we propose economic and eco-friendly strategies that could be harnessed sustainably and biotechnologically for resilient plant microbiome engineering.

## 1. Introduction

Plant microbiome engineering is claimed as an alternative emerged biotechnological approach for meeting the growing demand of foods under extreme climatic conditions. According to a recent survey by the United Nations (UN) (https://population.un.org/wpp/ (accessed on 20 September 2022)) and the United States Department of Agriculture (https://globalagriculturalproductivity.org/ (accessed on 20 September 2022)), the world population will surpass 9.8 billion people in 2050. To meet the population’s expected demand, agricultural productivity needs to be increased by up to 70% in food, fiber, and bioenergy. Green plants are the major and substantial source of food for every living organism on earth [1]. These plants are sessile and have been continuously exposed to the barrage of abiotic and biotic attackers, constantly struggling for their survival by evolving ample mechanisms to respond to and alleviate their negative impacts [2].

Unpredictable weather patterns and rising temperatures are among the effects of global climate change. These parameters have direct and indirect effects on the ecosystem and rhizosphere biology. Drought [3,4], salinity [5,6], osmotic and heavy metals [7,8,9,10,11], and high temperatures [12], are examples of abiotic stress that have detrimental impacts on agricultural yield. These stressors lower water absorption, nutrient acquisition [13,14], cause disease susceptibility [13], interrupt hormonal imbalance, and affect the plant’s photosynthetic efficiency. Certain environmental factors, i.e., temperature, CO_2_ concentration, and water availability, define the disease conditions of plants, which increase the susceptibility of plants [15,16,17]. According to the World Health Organization (WHO), some of the most important crops, including wheat, rice, oats, cotton, and barley, are vulnerable to numerous pathogens [18,19] when exposed to hostile environmental conditions. For instance, wheat susceptibility to *Fusarium graminearum* occurred on exposure to adverse environmental conditions. Biotic constraints are accounted for the detrimental elements of plant health and the agricultural economy. For instance, *Ralstonia solanacearum* and *Agrobacterium tumefaciens* are soil-borne bacteria that cause bacterial wilt and crown gall tumor, respectively, classified as second and third plant pathogens that cause serious plant diseases in various crop species worldwide [20]. Similarly, Dean et al. [21] classified *Fusarium oxysporum* Schlecht as the fifth most lethal and ubiquitously occurring soil-borne pathogen, causing vascular wilt in various plants.

Plant rhizosphere constitutes 1–3 mm of the soil around the plant roots, which provides an interaction site for microorganisms and plant roots. There is an enormous number of microorganisms that reside in the rhizosphere, colonize the different plant parts, and assist the host plant under adverse conditions. Microbial gene populations of the rhizosphere surpass plant cells population e.g., a one-gram root conserves about 10^11^ microbial cells population [22].

Both biotic and abiotic are the dominant factors that potentially influence plant microbiome compositions and functionality. Accelerating agricultural productivity to overcome food security using the available resources (land, germ plasma, water) without compromising the terrestrial, aquatic, and air ecosystem, is one of the biggest challenges that needs to be addressed and resolved.

In recent years, substantial efforts have been made to engineer the plant microbiome by deploying an array of highly advanced technologies and strategies, including rhizosphere microbiome (bacterial competitiveness engineering) [1], synthetic microbiome (genetically engineered microbes inoculation to host plant) [23] in situ microbiome (manipulation of the native microbial community in their native context) [24], and plant mycobiome (the optimization and improvement of beneficial plant-fungal interactions) [25]. However, on the other hand, all the potential biotic and abiotic perturbations that substantially influence or hinder their role are intentionally or unintentionally ignored.

Here, we explored the potential biotic and abiotic barriers that come in the way of plant microbiome engineering. Additionally, we spotted the limitations of molecular and biological techniques and the survival of biological organisms, which could be used in resilient plant microbiome engineering. Lastly, we propose and argue economic and eco-friendly strategies that could be harnessed sustainably and biotechnologically for resilient plant microbiome engineering. 

## 2. Plant Microbiome: Composition, Complexity and Diversity

Plants in close proximity with microbes that inhabit in soil are a complex medium, harboring a great number of diverse and versatile microbes, along with animals that hold at least a quarter of the total biodiversity of the earth [26]. Tens of millions of species of bacteria, archaea, fungi, viruses, and eukaryotes coexist underground, although only a few hundred thousand have been characterized in detail. This underground ecosystem serves as a repository for hundreds to thousands of millions of species of bacteria, archaea, fungi, viruses, nematodes, protists, etc., although some of them are explored taxonomically [27]. 

The rhizosphere is a hot spot for complex and bio-diverse micro (bacteria, fungi, oomycetes, viruses, protists, archaea) to macro (nematodes) organisms, considered a major engine of terrestrial biogeochemistry and performing an array of functions in the ecosystem. Rhizospheric soil, besides nurturing, enables microbial growth, therefore it is classified as mesotrophic [28]. This intricate narrow zone in the soil compartment is influenced by plant root exudates. A plant in the rhizosphere secrets a multitude of organic materials through rhizodeposition, that in turn attract microbes and therefore turns biodiverse, an enriched spot in the earth’s ecosystem [29]. Rhizodeposition is an active phenomenon that feeds microbial populations; hence this spot contains up to 10^11^ microbial cells per gram root [8] and houses more than 30,000 prokaryotic species, respectively [30]. 

Recently, some soil microbiologists have reported that a single gram of soil harbors hundreds to thousands of diverse microbial taxa—e.g., viruses, bacteria, fungi, oomycetes, and archaea. As per estimation, approximately 2.6 × 10^30^ prokaryotes dwelling in soil are thought to comprise separate genospecies (distinct taxonomic groups based on gene sequence analysis) in a range of 10^6^ to 10^8^. Soil also fosters an estimated population of viruses, and it is shown that per gram of soil harbors approximately 10^7^–10^9^ virus particles, which represents less than one phage per bacterial cell; this ratio, in comparison, is extremely small compared to that of the aquatic environment (Figure 1). Mendes et al. [30] holistically overviewed and portrayed the presence of microbial communities in the rhizosphere. This illustration explicitly measures the average number of genes in a given genome of representative species in the individual group of organisms. Studies have supported the complexity and enrichment of soil rhizosphere microbiome by reporting fungi/oomycetes 10^5^–10^6^ g^−1^, bacteria 10^8^–10^9^ g^−1^ archaea 10^7^–10^8^ (Figure 1) [29,30,31]. Most of the microorganisms in soil need special handling in order to grow in laboratory conditions and up to 99.9% remain uncultured [32,33,34,35]. Nematodes are the most ascendant and eminent bio-component of the soil ecosystem. Collectively, eukaryotes, nematodes, and fungi constitute a major portion of the soil. Their abundance is estimated at approximately four-fifths of all land animals. The densities of the nematode population are estimated at 10^1^–10^2^ g^−1^ in soil. Intriguingly, plant parasitic nematodes possess bipartite features and influence plant health and soil microbial population in two (positive-negative) ways in the soil ecosystem [36]. The term plant microbiome is vast, and it covers multiple associated components of above-ground (phyllosphere), underground (rhizosphere), and internal (endosphere) under its umbrella [37]. Zhang et al. [38] reported the population of microbes 10^5^–10^6^ g^−1^ CFU/cm^−2^ in the phyllosphere and 10^4^–10^5^ CFU g^−1^ in the endosphere. Plants are colonized by an astounding number of (micro) organisms that can reach cell densities much greater than the number of plant cells (Figure 1). Understanding all key factors that contribute to microbe-microbe, plant-microbe interactions, and microbe assembly, are still in their infancy. The complexity in interactive microbe-microbe and microbe-plant, and alteration in the microbial community composition at different phases of plant growth or in different plant tissues, is challenging regarding new insights in microbiome and engineering [23]. Underground communications in plants with a pool of good or bad microbes and animals are formidable [39]. The plant pathogen and plant-microbe interaction fields defined these terms. Beneficial microbes that support plant growth by helping plants absorb nutrients, fending off plant pathogens, and underpinning plants to withstand against biotic abiotic constraints, have been regarded are good microbes. On the other hand, plant parasitic fungi and nematodes are bad microbes, because they pose the greatest danger to plant health and inflict diseases on important crop plants that are economically significant and have a serious negative impact on food security. To date, no study is reported to reveal plant response to select beneficial microbes and abstaining harmful microbes in plant microbiome interactions.

## 3. The Dilemma of Plant Microbiome Interactions

The rhizosphere microbiome is an intricate, diverse, and complex microbial zone that nurtures bacteria, fungi, oomycetes, viruses, and nematodes. This diverse array of organisms inhabiting the same niches, ecologically influence each other in terms of functionality and diversity, exerting manifold complex mechanisms. For instance, these organisms always struggle for survival, progeny replication, food, space, and colonization in the plant rhizosphere. This hotspot regime (rhizosphere) possesses beneficial (plant growth promoting) and harmful (plant pathogens) organisms that concomitantly confront plants with no additional choices. Rhizosphere microbial and animal populations influence each other as well as attributing (positive or negative) to plant health. Moreover, they establish a mutually (non)beneficial relationship, where plants either get healthy and protected from unwanted invaders or are eventually diseased [40,41]. 

Additionally, such a complex interaction impacts plant health and fitness. To this end, the related microbiome assists the plant in taking up nutrients (e.g., P, Zn, K, etc.) for their solubilization and becoming stimulated through direct collaboration with the microbially biosynthetic phytohormones (e.g., auxins, gibberellins, cytokines, etc.) [42]. These strategies have been called direct mechanisms in plant growth stimulation [43,44]. Phenazines are nitrogenous heterotricyclic metabolites, usually produced by bacteria fluorescent pseudomonads in plant root microbiomes [45]. Such metabolites are active protectors; for example, wheat and tomato crops and from pathogenic fungi and water molds [46,47].

Soil-borne pathogens also dwell in the rhizosphere microbiome and are considered integral components in plant-microbes composition, structure, and communication. Along colonization with the host, pathogens secrete effector proteins to promote disease elaboration under different mechanisms. In a successful colonization and infection, soil-borne pathogens deter plant physiology and hinder its growth that subsequently reduces plant productivity. Therefore, about 20–30% loss in agriculture production is estimated, prominently due to stated pathogens around the world [34,35]. 

Although studies claimed that plants have sensing capability towards the selection of favorable microbes in the bulk soil [48], in contrast, plants still face a confounded condition within microbiome recruitment and proper selection. For example, Zipfel et al. [49] demonstrated that both rhizobia and soil-borne pathogens concomitantly infect *Medicago truncatula,* and thus it is unable to refrain pathogens from colonization.

It is undeniable that plants use complex defense mechanisms to fend off attacks from phytopathogens and endure less suffering, including provoking innate immunity and creating multilayered physical barriers [50]. Inevitably, phytopathogens hijack the plant’s innate immunity, then colonize and potentially infect it by evolving a strategic invasion that damages the plant roots and seriously causes devastating disease [51]. 

The application of biological control is a highly recommended and well-known strategy in plant disease management. In the last few decades, this eco-friendly and sustainable strategy has gained massive attention in the agriculture sector [52]. Recently, by using biological control agents (BCAs), researchers have proposed multifaceted approaches for microbiome engineering in which these agents could be harnessed to suppress the activities of phytopathogens and underpin the beneficial functions of the microbiome for plant health. These approaches, such as Host Mediated Microbiome [53], Synthetic Microbiome [23], Artificial Seed Microbiome [54], In Situ Microbiome [24], and Plant Mycobiome [25], are regarded as top listed in the subject of interest. 

Liu et al. [55] have deduced that root-associated plant growth promoting bacterium, such as *Pseudomonas* spp., elicited induced systemic susceptibility (ISS) rather than inducing systemic resistance (ISR) to foliar pathogens in plants. This is a shocking and alarming call for biocontrol agents and it might compel scientists to think harder before suggesting BCAs for plant disease remedy and plant microbiome engineering.

In the same way, Snelders et al. [56] reported that plant pathogens evolve a sophisticated mechanism to manipulate host rhizosphere microbiome for their advantages in turn by the identified virulence effector VdAve1, secreted by the fungal plant pathogen *Verticillium dahlia*. This effector (VdAve1) displays antimicrobial activity and suppresses antagonistic bacteria of the host-microbiome resulting in facilitating colonization of tomatoes and cotton. This study demonstrates that a fungal plant pathogen based on effector proteins modulates microbiome compositions in or outside the host, and proposes that pathogen effector catalogues represent an untapped resource for new antibiotics. Besides the known activities of plant pathogen effector proteins in targeting host physiology, including immune responses, it is also used in self-defense against host-secreted defense molecules. These effectors also suppress the activities of beneficial microbes in the host-microbiome, and consequently reduce microbiome resilience and functionality.

Interestingly, the pathogenic fungus, *Fusarium oxysporum* F. spp. radicis-lycopersici has been shown to colonize similar zones in the tomato root as plant-growth promoting phz+ pseudomonads, where they compete for plant-root exudates [57]. The proposal of microbes engineering itself confronts a variety of challenges, e.g., narrow persistence, gene transformation between microbial strains in soil microbiome, competition for niches and food, and environmental issues. How does the native microbiota respond to the alien microbial strain upon introduction? To date, there is a lack of knowledge or research gap to show the underground soil ecosystem. For instance, maybe the endemic AMF communities are nonfunctional due to the introduction of commercial AM inoculants, and lose the naturally beneficial AMF communities [58].

Amongst environmental and climatic elements, soil, geography, and farming activities, as well as plant domestication, could differentiate plant-relevant microbial communities [59]. Changes in environmental components can result in plant phenotypic modification, as previously reported by Valladares et al. [60], and consequently can alter the assemblage of distinct microbiomes harboring plant compartments [61]. Various soil factors viz. soil types, pH, carbon nitrogen ratio (C/N) ratio, as well as the availability of phosphorus (P) and potassium (K) in soil, have been common determinants in root microbial community composition by affecting plant growth and immunity. Dombrowski et al. [62] sampled arctic-alpine Arabis alpine from native locations, grown under controlled conditions, that allowed the examination of root microbiome using 16S rRNA amplicon sequencing. From such analysis, the soil type and duration were fundamental drivers that caused approximately 15% modification in the root microbiota. In addition, soil pH and carbon, C/N, water content and biogeography have been shown to be highly influential to the composition of the microbiota [63]. Amongst these environmental factors, the soil pH strongly impacts the composition of the total community at a very high taxonomic level [64]. Yet a decrease in microbial activities and communities has been also reported, due to heat disturbance caused by natural wildfires [65]. Moreover, an increased contamination level of petroleum products can cause modifications such as in the structure of the willow microbiome. These alterations can be less extreme to the rhizosphere and plant tissues but can be effective to bulk soil. It is explored that plants, by providing a more precise condition, can protect microbes against any contamination [66]. Climate directly affects plant and soil microbiome, yet to date, its role has largely been unheeded, or of little interest [67]. However, studies conducted for Britain ecosystems have found rainfall and temperature have decisively attributed to the structural composition of plants’ bacterial community [68].

Recently introduced high throughput “omics” approaches have exclusively portrayed microbial’s functions and molecular mechanisms that occurred in the complex interactions of plant microbiomes. Most beneficial microbes are closely related to pathogens; however, it is unclear how the plant immune system distinguishes these microbes to contest infection or facilitates colonization. To respond to this question, certain fundamental approaches can be highly useful. These incorporate unraveling the genetic microbes and metabolic relationships, such as signaling events that mediate microbe-host interactions, along with the inclusive quantifiable systems biology approaches [69]. Plant microbiome engineering has been portrayed as effortless and fascinating in numerous studies. Its manipulation and application in the field condition are intractable [70]. Unfortunately, unintentionally, a range of studies have overlooked potential abiotic barriers, confronting plant microbiome engineering. Here, we tried to spot all effectual components being exploited in microbiome engineering, yet they are the main challenges in plant microbiome engineering. 

## 4. Deciphering the Potential Role of Micro-Macro Fauna in Plant Microbiome Engineering

Rhizosphere microbiomes nurture diverse, complex, and dynamic microbial communities executing vital ecosystem processes, thus characterizing the hot spots of the earth [1]. Plants and microbes are coevolved and the significant of plant-associated microorganisms in plant growth, development, nutrient acquisition, protection and increase tolerance against biotic and abiotic stresses [8,71,72,73,74,75,76] are unquestioned and has been well studied under the concept of holobiont [1,77]. Root surfaces tightly adhere to the rhizosphere’s soil interface that harbors these plant growth-promoting rhizobacteria (PGPR) [28,78]. These “PGPR”, can concurrently perform the functions of nitrogen fixation, nutrient mobilization, growth regulator synthesis, and disease suppression, to promote soil microbial production, nutrient availability, and plant productivity [79,80], as depicted in Figure 2. Plant growth-promoting rhizobacteria -mediated biocontrol processes are wide-ranging, such as nutrient availability and ecological niches, synthesis of allelochemicals including enzymes and antibiotics, induced resistance against plant pathogens, as well as the promotion of helpful microorganisms (Figure 2) [81]. 

The best-known PGPR colonizing in the rhizosphere strains incorporated *Bacillus, Rhizobium, Acinetobacter, Alcaligenes, Azotobacter, Arthrobacter, Enterobacter, Pseudomonas, Serratia,* and *Burkholderia,* and successfully induce disease resistance against bacterial plant pathogens, such as *R. solanacearum* [82], *E. carotovora* [83], *D. solani*, *E. amylovora,* and *P. carotovorum* [84]. 

Both growth promotion and biological control can be regulated by the same PGPR and endophytes strains and these traits are revealed by their genome studies [85]. Generally, the biological control of these bacteria relies on direct or indirect modes of action; however, all these mechanisms are highly influenced by the type of host plant [86,87,88]. In direct mechanism, pathogens directly affected by the production of metabolites, for instance, antibiotics, hydrogen cyanide (HCN), iron-chelating siderophores, pyoluteorin, tensin, 2,4-diacetylphloroglucinol, phenazines, viscosinamide, and other cell wall-degrading enzymes, while another known mechanism is induced by systemic resistance, happening by the intervention of inducing agents that systemically stimulate chemical or physical defensive mechanisms of the host by resulting in the decreased symptoms of pathogens that invade tissues distal to the inducer (Figure 2) [89,90]. Root-associated microorganisms can influence plant hormonal balance in several ways. Rhizospheric microorganisms can produce or degrade the given hormones of ethylene, auxin, cytokinin or gibberellin, with broad repercussions on plant phenotype and fitness [91]. Microorganisms in the rhizosphere can also induce systemic resistance against plant pathogens. This priming prepares the plants to produce stronger and faster responses to future exposures [92]. Fungal communities are integral components of plant microbiomes that assist growth and plant physiology. Plant associated fungi establish symbiosis relationship with plants, for example, AM fungi in land plants that were found around 300 million years ago [37,93]. This mutual relation represents one of the beneficial interactions on earth because most plant species continue to take advantage of this mutualism. Plants are highly dependent on AM fungi to collect, and transfer, phosphorus or other nutrients that limit plant growth, in exchange for carbohydrates. Plant roots are knowingly infected by AM fungi that send a threadlike hypha beyond the root zone to mining nutrients from the soil, and transport them back to the host [94,95]. 

Plant associated fungal communities stupendously serve as a natural enemy of plant pathogens. These are biologically controlled plants from pathogenic infections in the natural ecosystem. The fungal biological controlling agent uses various mechanisms to hamper and suppress pathogenic activities. For example, *Trichoderma* is the most frequently commercialized biocontrol agent that evaded various plant diseases to date [96]. The species of the *Trichoderma* infects several plant pathogens and oomycetes. Within the growing rhizosphere, this spp. outcompetes the related pathogens and secretes plant growth promoters [97]. Several useful fungi are well-known hunters of plant parasitic nematodes. For instance, *A. avenae* reduces plant-parasitic nematode *Ditylenchus* propagation, which has been a biocontrolling active agent in certain pathogenic fungi or parasitic nematodes contexts [98]. The mycorrhizal fungi can colonize host plant roots, and through inducing systemic resistance (ISR) can protect the plant from a range of parasitic nematodes [99,100]. Many studies have been conducted in laboratories under greenhouse settings, whereas their efficacies in field conditions are lacking. Transforming all laboratory successes into equally effective field applications can represent the next step of the challenge.

Protists are representatively paraphyletic and unicellular eukaryotic prominent groups that include a majority of eukaryotic phylogenetic diversity [101]. Due to a closer susceptibility to fungal or bacterial communities, such groups are abundantly dispersed in the plant rhizosphere. Whereas, they substantially contribute to production, food web interactions, and nutrient recycling. They also represent diverse moods of nutrition, such as phagotrophy, phototrophy, symbiosis, saprotrophy, or a combination of these strategies, in a given ecosystem [102,103].

Protists-prey interactions are varied, including trophic interactions and chemical communication (Figure 2). In turn, these interactions can cause important changes in microbiome structure and activity. Predation typically decreases total bacterial biomass [104]. Along with the increase in nutrient turnover, protist predation stimulates microbial activity, which is evidenced by enlarged microbial respiration and nutrient mineralization [105]. Predation results in abundant organisms harboring traits conferring resistance to protists [106]. They also stimulate trait expression linked to defense [107]. These antipredator traits are highly useful in the delivery of microbiome functions relevant to plant health. Nevertheless, several secondary metabolites conferring resistance against consumption by protists are also involved in suppressing plant pathogens and immunity [106]. Protists can impact the effective microorganisms on plant hormonal balance by altering their abundance and activities. For example, by promoting auxin-producing bacteria [108] they can trigger lateral root branching. Protists also increase the level of plant cytokinin, possibly as a result of increasing nitrate concentration that occurs when excess nitrogen is secreted [104]. Finally, protists indirectly alter plant hormonal balance by affecting microbiome functions. The cumulative production of bacterial metabolites, such as 2,4-diacetylphloroglucinol (DAPG) [107], is an antimicrobial compound that interferes with auxin signaling [109]. A strong effect of protists on plant metabolome can most likely be linked to these multiple hormonal changes, respectively [110].

Earthworms are fundamental engineers in the soil ecosystem. They perform the main functions in soil structure. However, there is limited knowledge available on how earthworms represent long term effects on soil microbiome. They are the most typical soil rhizospheric inhabitants [111], contributing to a great part of the biomass (>80%) in the soil macrofauna [112]. They interact with the soil microbiota, termed “Sleeping Beauty Paradox” [113]. The earthworm effect can be neutral, negative, or positive on the richness and diversity of microbial communities, depending on earthworm types and on the “micro-habitat”. Field and laboratory studies have revealed major connections between earthworms and microorganisms that increase carbon turnover, nutrient availability, and microbial activity in the soil [114].

Besides a comparative stimulation of the prokaryotic community in the eukaryotes, earthworms increase microbial resilience in community co-occurrence networks [115]. Some soil microorganisms remain in the dormant stage due to unfavorable environmental conditions. Earthworms produce and release a cutaneous mucus, known as glycoproteins, present in the rhizosphere, and activating activities of microorganisms and bridge interactions. According to Bedano et al. [116], earthworms are principal ecological mediators that significantly influence soil biological functions and microbial activities. This occurs because earthworms release energy-rich mucus molecules that along with supporting, enhance microorganisms’ activities under the “priming effect” [117]. Additionally, they generate molecules that effectively behave like hormones and mediate signals into the plant genome and ultimately mediate gene expression [118]. *Rhizoctonia solani* is a damaging soil-borne plant pathogen, which has principally damaged cauliflower seedlings called damping off, also controlled by *A. avenae* and *Aphelenchoides* spp. [119].

This argument has been proven by Elmer et al. [120], where it was found that along with the augmentation of earthworms, the population of the *fluorescent pseudomonads* and filamentous actinomycetes proliferate in the soil rhizosphere. Stephens et al. [121] have also tested *Verticillium wilt* with eggplant to see the effect of earthworms on microbial populations, where earthworms have found significantly increased population densities of *Bacilli* and *Trichoderma* spp. as compared to uninoculated soil.

Microorganisms have been found digested by earthworms [122], thereby declining microbial biomass, in particular fungi [123]. They have selected or stimulating roles in soil microbes [124] that help them digest soil organic matter, since the earthworm gut often lacks adequate functional enzymes [125]. The related process can improve the soil in certain bacterial taxa. In line, bacteria capably decompose organic matter that is favorably consumed by earthworms, or in denitrifying bacteria able to survive under low oxygen conditions of the earthworm gut [126]. McLean et al. [127] reported some contradictions by exposing that earthworm invasion tends to decrease fungal species, or their diversity and richness. Earthworm presence decreased zygomycete species considerably, due to disruptions in the fungal hyphae. The presence of earthworms can somehow physically interrupt hyphae that can further justify mycorrhizal decrease, or their morphological alterations.

## 5. Omics Tools: Putative Role and Limitations in Plant Microbiome Interactions

The omics technologies capably help to screen and combine the epigenomics, transcriptomics, proteomics, and metabolomics information in plants. With the advent of modern analytical paradigms, the data attained through any of the omics approaches could be linked with the microbiome of interest. Progress in cutting-edge next-generation sequencing, with decreased costs, have permitted to acquire plant genotype through single nucleotide polymorphisms (SNPs). In future, the data attained through shotgun metagenomics will promote mapping microbial functions, as recently conducted in the phyllosphere microbiome of rice [128].

The metagenomic, metatranscriptomic, and further functional screening as proposed, can allow instrumental investigation in any complex ecosystems of host organisms, biogeochemical (interactions in soil/rhizosphere/plants), pathogenic, biochemical, and metabolism, as well as their mutual exchanges. Omics-based generated details facilitate tools to respond, intrinsically arising fundamental questions in a microbial population. Even failure in metagenomics techniques cannot differentiate Relic DNA along sequencing of the targeted sample. While various studies have claimed that plants can capably sense the selection of useful microbes in bulk soil [48], in contrast, to be able to select or recruit beneficial microbes, plants still face confounded conditions within the respective microbiome, while molecular mechanisms underlying such recruitment are yet unraveled.

The meta-omic analysis aims to detect microorganisms, genes, variants, metabolic pathways, and or ecological functions that can illustrate microbial community in the sample of interest [129]; none of the bacterial activities, roles or interactions of these with further species have been reported. Metatranscriptomics (c-DNA sequencing) coupled to metaproteomics and metabolomics has exclusively described the entire community covered through the metagenomics. Metagenomic, along with the metatranscriptomic, has fully investigated the genomic composition as well as the diversity across communities by molecular culture independent sequencing methods [130,131]. This included the targeted sequencing of bacterial 16S rDNA, eukaryotic 18S rDNA, fungal ITS (Internal Transcribed Spacer) [132], as well as whole-metagenome shotgun (WMS) sequencing. The combinatorial multi-omics data resulted from metagenomic, metatranscriptomic, metaproteomic, and metabolomic, have previously been implemented, in order to understand soil microbiome and associated alterations in the molecular contents on community level induced by ecological interactions [133].

The variety of microbiomes can be identified using metagenomic methods, and “-omics” methodologies (genomics, proteomics, and metabolomics) and can be used to investigate how genomic information is converted into structures and functions in the interactions between plants and their microbial partners. However, low RNA contents and its low stability in soil and the information regarding the enzyme translation are the potential limitations of next generation sequencing (NGS). Studies have shown that the transcription of genes encoded hormone regulation, secondary metabolism, and stress response, were upregulated in plants treated with plant growth promoting bacteria (PGPB). Firmly established rhizospheric PGPB rely upon several chemical and cellular factors. These factors can incorporate the induced mechanism against multiple stimuli, metabolic versatility, biofilm formation, and release secondary metabolites that maintain a cross-talk between bacterial cells and host plants [13,134,135,136]. In line with Zhang [50], we have studied an entire rhizospheric genome in the *Brevibacterium frigoritolerans* (ZB201705) from maize under drought or salt conditions. These authors have observed that the given strain capably generates manifold proteins. However, mechanistic insights can possibly be tracked through the proteomics approach for the produced proteins. The application of proteomics in such a scenario is highly useful because it can provide a thorough understanding for the mutual association of bacterial community with that of the host plant [137]. Based on the metaproteomics, proteins could be used to recognize how microbes attribute in the modification of the soil ecosystem, providing details about enzymes secretion and the microbes that exert metabolic capabilities in soil. A small proportion of proteins in the soil sample can be challenging because of (i) the presence of a small amount of protein, and (ii) soil chemical contamination that make protein extraction or purification critical. Metaproteomics can holistically portray all active members in bacterial population, as observed when a vineyard was subjected in a cohesive pest management [131], where metabarcoding was used in the characterization of bacteriome [138].

Throughout, metabolomics is another tool that could be used for the characterization of metabolites [139,140,141]. Overall, in such metabolites, many molecules can be highly interesting that are used by organisms of mutual communications, e.g., signaling, defense and many other mechanisms [142]. Most of these molecules are of prime interest because they are the basis to carry out manifold communication amongst social insects (ants and bees) and plants in general. These chemicals are produced as sensory molecules by the host organisms when exposed to some external stimuli, for example, herbivores, pathogens, etc. In line with the technological advancement, these communicatory metabolites are possibly being profiled by metabolomics, such as the metabolomics profiling. Metabolomics profiling is an emerging tool, commonly used to determine the degree of changes in the concentration of predefined metabolites that can occur in response to external stimuli [143]. These stimuli can be classified as the environmental adaptation, ecological factors, drought and salinity, or further modification in a given rhizosphere metabolome by other means. However, many modulations that have occurred in the rhizosphere metabolome are already been captured by metabolomic that additionally uncovered root exudating metabolites, especially the aromatic organic acids [89,144]. The advanced molecular techniques still face difficulties to attribute the species that produce the identified metabolite. How to identify, purify, and quantify the metabolites present in soil are potential bottlenecks of omics tools.

## 6. Disease Suppressive Soil: Underlying Mechanism and Manipulation for Healthy Microbiome Engineering

Soil is the composition of unconsolidated mineral or organic material, which carries out numerous biogeochemical cycling in the ecosystem [145]. This dynamic and complex ecosystem hosts billions of microorganisms and microbiomes, which provide food for all living organisms [146]. One teaspoon full of healthy soil fosters between 100 million and 1 billion organisms alone [147]. Soil is composed of biological and physical elements; collectively, they perform a vital role in soil chemistry and physiology. It harbors diverse groups of microbial and animal populations including bacteria, archaea, fungi, viruses, algae, protozoa, and nematodes (Figure 3) [32]. These macro and microorganisms drive substantial ecological processes that execute primary services in plant health and productivity [148]. The characteristics, functions, and interplay with plants define the status of the soil. Soil disease suppression is a state of the soil, that originally originates from the diverse microbial communities. Generally, disease suppressive soil defines that a type of soil inhibiting the plant pathogens and inoculum, but they are not capable to establish or persist, and cause little or no disease [149]. The microbial community structure, composition, assembly and functionality play a vital role in the development of disease suppression against soil-borne diseases [150]. Disease suppression establishes by the collective contributions of all bio-diverse microbial communities of the soil ecosystem (Figure 3).

Bio-diverse microbial communities dwelling in soil deploy multifaceted mechanisms to inhibit or reduce the soil-borne pathogens activities. The attenuation of pathogenicity of the soil-borne pathogens occurs due to the employment of some specific functions, such as antibiosis, parasitism, competition for resources, and the predation of beneficial biological control microbes in disease suppressive soil (Figure 3).

Although the crucial role of abiotic factors in soil suppressiveness against plant pathogens could not be overlooked, here, we are only concerned and focus on the contribution of biotic elements in disease suppressive soil. In microarray analysis, non-pathogenic microbes such as *Streptomyces* (Actinobacteria), *Bradyrhizobium*, *Burkholderia,* and Nitrospira, were observed to be highly active and carry out more functions in suppressive soil [151]. Interestingly, the beneficial plant growth promoting soil microbes are not the only causing factor of soil disease suppression. This argument was defended by Durán et al., 2018 [152], in their study that took all the diseases of wheat (*Gaeumannomyces graminis* var. Tritici), mediated through bacterial endophytes rather than rhizospheric microorganisms. This study also recommended that only pathogen biomass reduction is not determined by the disease suppression. The diversity of the fungal community is also linked with higher disease suppression [153]. To understand the underlined mechanisms of soil suppression against soil-borne disease, it is crucially important to decipher the role of biotic and abiotic factors contributing to soil suppressiveness. It is worth noting to give intention to the complex interaction between plant and associated microbiota, and the influence of environmental factors on disease suppression in the soil ecosystem.

Some studies have documented the antagonistic microbes that contribute to the biocontrol mechanisms of the involved nematodes detected mostly in suppressive soils. These microbes applied as consortia, such as the fungi *Pochonia, Dactylella, Nematophthora, Purpureocillium, Trichoderma, Hirsutella, Haptocillium, Catenaria, Arthrobotrys, Dactylellina, Drechslerella*, and *Mortierella*, and the bacteria *Pasteuria, Bacillus, Pseudomonas, Rhizobium, Streptomyces, Arthrobacter, Lysobacter*, and *Variovorax* [119,154,155]. The bacterial antagonists often have more than one mode of action. The soil suppressiveness is critically sensitive to biotic and abiotic factors and its disease suppression is completely dependent on these attributes. Soil fertility and physical traits, biodiverse microbial populations, and agricultural management practices highly affect the degree of suppressiveness of the soil ecosystem [156]. Again, here we will shed light on the biotic components linked to soil suppressiveness. The manipulation of suppressive soil with a beneficial biological control agent will result in either enhancing, reducing, or eradicating disease suppression.

## 7. Sustainable Approaches

Organic Fertilization

Organic amendments (OAs) have been a major practice in traditional and organic agriculture. The addition of OAs, beyond enriching soil structure and fertility [99,157], can help in protecting crops from diseases that are generally produced by soil-borne pathogens [158]. Plants and derived products incorporating OA, crop residues, compost, manures, fish and blood meal, biochar, and chitosan, etc., have significantly reduced the occurrence rate of soil-borne diseases [30]. Within these products, organic matter has a great role in the soil because along with the composting or decaying, such matter can boost soil fertility, health, and feed the microbes of the soil microbiota [159]. Compost-inhabiting microbes generate hormones useful in plant growth and other antagonistic metabolites (siderophores, tannins, phenols, etc.) against soil-borne pathogens. Manifold benefits have been reported for example, biochar has a central role in soil health, plant growth, nutrient availability, water conservation, harmful elements manipulation, alteration/transition in microbial community size, prevent nutrients leaching, and subsequently leads to soil fertility with the reduction of fertilizer use by supporting a sustainable environment [160,161,162,163]. 

Disease suppression can improve plant growth and strength induced by the nutrient’s accessibility, the existence of the useful microbes, and well-drainage system [164]. The decomposition of the OAs in soil can produce many nematotoxic or nematostatic volatile organic compounds (VOCs). Several such chemicals are commercially developed nematicides, such as methyl isothiocyanates (MITCs). MITCs are derived from various plants in the Brassicaceae family, such as *Brassica juncea*, *Brassica napus* and *Sinapis alba.* The addition of MITCs as amendments into the soil have represented greater in vitro activities against parasitic nematodes [158,165,166]. Tissue exudates from *Crotalaria juncea* demonstrated allelopathic alkaloids, e.g., the monocrotaline and pyrrolizidine of natural nematotoxic behavior [167]. Amongst OAs, chitin, pine bark, and chicken litter are the major contents, providing an enhanced nematode antagonistic microflora in the soil [168]. Culbreath et al., 1985, [169] have found that subjecting 1% chitin has greatly promoted microbes in the soil. This represents an established enzymatic chitinolytic relation between mycofloral and the parasitized nematode eggs that contain chitins in a favorable amount. Further studies based on OAs strategies, have found that adding 5% (*w*/*w*) of composted pine’s bark powder in the soil has proven better resistivity against nematodes of root-knot (*Meloidogyne arenaria*) and soybean cyst (*Heterodera glycines*) [169,170]. Plants are principal producers of diverse vital metabolites, such as the essential oils that contain terpenoids, phenols, alcohols, organic acids, and more compounds of biocidal nature.

OAs have multiple functions, including disease control in plants. Together with promoting the activities of newly introduced biocontrol agents, they can support soil microbiota by naturally suppressing plant pathogens [171]. Within an array of molecules in OAs, some compounds capably induce plant resistivity in unfavorable circumstances [172]. Other than the microbial effect of chitin on nematode infections, plant chitinase induction has been observed with enhanced resistivity towards detrimental microbes [173]. Some reliable controlling observations for *Verticillium dahliae* [174], *Thielaviopsis basicola* [175], different species from *Fusarium*, *Phytophthora* [149,176] and *Pythium* have been established. The strategy of using OAs can be highly efficient in the agroecosystem, where its implementation can control soil-borne pathogenically caused diseases to a better plant production. Further studies are of high relevance to understand the mechanism(s) in suppressive and conducive effects, identifying different types of OAs, and application timing that can lead to optimize plant productivity and disease control in varied agroecosystems.

## 8. Biological Fertilization

The uncontrolled application of synthetic fertilizers to feed the world’s growing population has undoubtedly contaminated the environment and seriously damaged beneficial insects’ habitats. However, utilizing too many chemical inputs has diminished soil fertility and increased the crop’s susceptibility to diseases [177]. Due to the potentially harmful impacts of chemical fertilizers, biofertilizers are meant to be a safe substitute for chemical inputs that significantly reduce ecological disturbance. Biofertilizers are economical, eco-friendly, and significantly increase soil fertility, when employed for an extended period of time [178,179]. The process of adding biological amendments to soil with the goal of enhancing the effectiveness of the soil microbiome is referred as “biofertilization”. [102]. Due to its broad potential in enhancing crop productivity and food safety [180], the use of microorganisms as biofertilizers is somewhat regarded as a substitute for chemical fertilizers in the agricultural industry. In the agriculture sector, it has been observed that some microorganisms, such as bacteria, fungi, and cyanobacteria that promote plant development, have demonstrated behaviors similar to those of biofertilizers. Numerous studies on biofertilizers have shown that they can provide the crop with the necessary nutrients in appropriate proportions, which has increased agricultural production [181]. For instance, various Rhizobia on plant growth and biocontrol, such as Rhizobium, *Bradyrhizobium, Sinorhizobium, Azospirillum, Nostoc, Anabaena, Acetobacter, Bacillus megaterium, Azolla,* and *Bacillus polymyxa,* etc., are very common plant growth-promoting bacteria (PGPB) that aid in and significantly increase crop yield and overall plant growth. Some of them are plant nodule-residing symbionts, free-living nitrogen-fixing bacteria such as *Azospirillum*, *Azotobacter*, *Rhizobium, Pseudomonas*, and *Burkholderia* and play an important role as biofortifying agents. Numerous soil bacteria act to solubilize the mineral phosphate in the soil [102,182], such as the genera *Bacillus*, *Rhizobium*, *Pseudomonas*, *Burkholderia*, and *Enterobacter* [183].

Biological fertilization offers better nutrient conditions for the accumulation of ferulic acid, p-coumaric acid, isoquercitrin, and quercetrol than organic fertilization, where rutoside accumulates at the highest rates [184]. Another benefit of biofertilization is that it helps the soil become more resistant to diseases. By increasing the quantity of native microorganisms with antifungal activity, a diseased soil treated with biofertilizer based on Bacillus and Trichoderma spp. was able to reduce the *Fusarium oxysporum* pathogen [185]. It has been proposed that bio-organic additions could help maintain a comparatively stable rhizosphere microbiome by replacing at least 25% of synthetic fertilizer consumption [186].

## 9. Conclusions and Future Remarks

Currently, a supreme challenge around the world is food security. Chemical pesticides and fertilizers have been utilized by agricultural platforms for a very long time. Utilizing such resources aims to improve food output in order to meet the demands of the growing human population. Employing these organic pesticides and fertilizers in excess may not be the best option for maintaining sustainable ecosystems. The major goal in the present agricultural practices is to amplify eco-friendly food, fiber, and fuel on limited land, using fewer agrochemicals and fertilizers without being compromised the ecosystem and human health. Plant microbiome engineering thus emerged as an alternative untapped source that could be exploited for plant growth improvement and high-yield production. However, plant-microbe interaction is still a dilemma for plant immunity, thus it is unable to discriminate between friends and foes in the underground ecosystem of the soil. This is now considered one of the hot questions in the field of plant-microbiome interactions and microbiome engineering for plant microbiologists, ecologists, and plant breeders, respectively. 

Extensive research and a review of the literature has been reported in the last decade, proposing microbiome engineering for plant growth and sustainable crop production in field conditions; but the significant challenges of plant microbiome and its engineering are almost ignored. The inoculation of PGPRs into the rhizosphere undoubtedly achieves those respective goals, yet, it needs to dig deeper into rhizospheric microbial ecology to understand the interaction between exogenously inoculated microbes, existing microbiota, and host plants, using omics technology. On the other hand, miraculously, the discovery, screening and advent of biological control agents into agriculture sciences provided massive relief to human and environmental health, and contributed highly to sustainable and eco-friendly agriculture production. As the potential role of micro-macro fauna in plant microbiome engineering is scarcely investigated, further efforts are required to monitor and decipher its interaction with the host and its exploitation for microbiome engineering. Additionally, metagenomics studies are rapidly uncovering the compositional richness of microbial communities in diverse habitats, however, the limitations of omics tools in plant microbiome studies needed to be addressed in order to reveal insights into plant microbiome interactions. 

Sustainable approaches and biofertilization optimize the plant microbiome interaction and functionality, which could be harnessed for resilient plant microbiome engineering. Organic amendments feed on soil microbes, and can explicitly support their nourishment in a given rhizosphere, which improves plant production. However, the soil-borne pathogens and pests also nourish the same food, resulting in an increase in their population, gaining strength against plant-associated beneficial microbes. All of these barriers need to be considered in order to engineer an optimized plant microbiome. To engineer the fully optimized rhizosphere microbial interaction, synthetic biology, omics biotechnology, and the recently emerged genome editing techniques might be capable of understanding the rhizosphere microbial interaction, and exploiting it for plant health and disease management to attain the goal of zero hunger for continuously growing population in sustainable agriculture.

## Figures and Tables

**Figure 1 biology-11-01782-f001:**
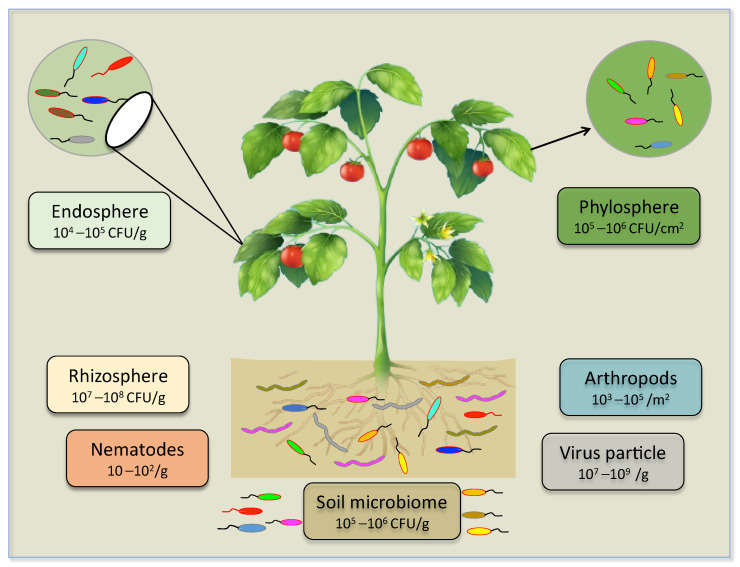
A holistic overview of plant microbiome composition, complexity, and diversity.

**Figure 2 biology-11-01782-f002:**
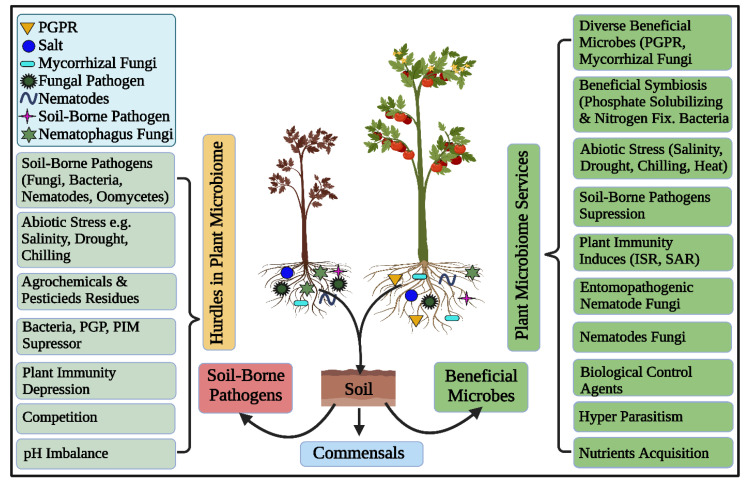
Schematic diagram showing the potential services and hurdles of the plant microbiome. The host plant encounters both beneficial and harmful components concomitantly. The below-ground complex ecosystem harbors biotic and abiotic compositions that significantly influence plant health and fitness. The host plant thrives in a healthy microbiome, contrary, it gets diseases in an unhealthy and imbalanced microbiome.

**Figure 3 biology-11-01782-f003:**
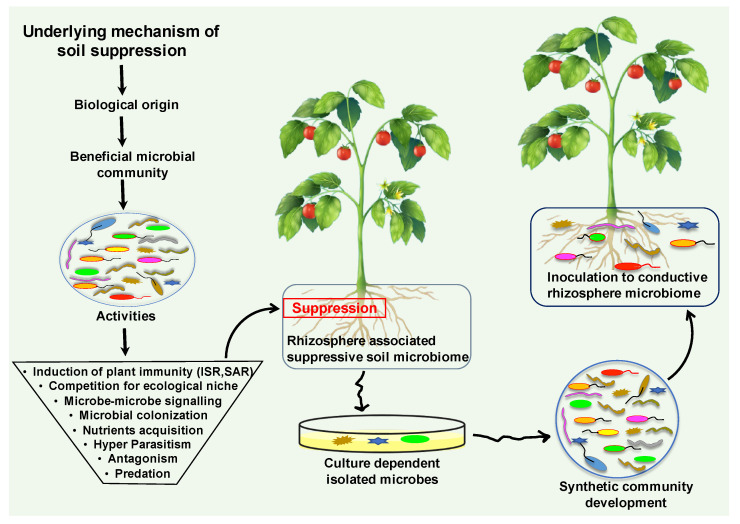
The diagrammatic illustration of suppressive soil microbiome origination and the underlying mechanism. The microbiota establishes the disease suppression soil microbiome by employing an array of direct and indirect mechanisms. Collectively, these diverse microbial communities perform significant beneficial activities that promote plant growth and suppress harmful pathogens functions. Suppressive soil microbiome could be harnessed for resilient microbiome engineering via screening and isolating the responsible candidates of disease suppressive soil. Development of synthetic communities from suppressive soil microbiome could be inoculated into a conducive soil microbiome against soil-borne disease.

## Data Availability

Not applicable.

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
