# Peer review of "Plant Microbiome Engineering: Hopes or Hypes"

_biology, 2022, doi:10.3390/biology11121782_

Round 1
Reviewer 1 Report
The review by Afridi et al. (biology-203419) “Plant Microbiome Engineering: Hopes or Hypes” makes an overview of plant microbiome composition, and describes the recent developments and achievements made around the challenges in plant microbiome engineering. The authors provide relevant information about this field here, however, this topic has been widely covered in science over the last few years. I, thereupon, wonder about the impact that it could have on the readers of /Biology/ since there are already many Reviews published about it.
I hope that my comments can help to get an improved version:
Major comments:
- The abstract is too long and does not highlight the relevance and impact of this manuscript to be published. I encourage the authors to rewrite it and use shorter sentences for better understanding. Especially lines 30-45 appear repetitive and unclear.
o Lines 30-32: do not clarify the meaning of Rhizosphere microbiome
o Lines 32-34: please, avoid subjective statements.
o Line 34: which argument?
- I found the structure of the Review quite unbalanced. Most of the manuscript is an Introduction, and the rest (2. Conventional approaches; 3. Conclusion and future remarks) is vaguely described and not well covered. What does this review offer that the others already published on this topic do not offer?
- I suggest the authors include the following published Reviews in this manuscript:
o Lau, SE., Teo, W.F.A., Teoh, E.Y. et al. Microbiome engineering and plant biostimulants for sustainable crop improvement and mitigation of biotic and abiotic stresses. Discov Food 2, 9 (2022). https://doi.org/10.1007/s44187-022-00009-5
o Albright, M.B.N., Louca, S., Winkler, D.E. et al. Solutions in microbiome engineering: prioritizing barriers to organism establishment. ISME J 16, 331–338 (2022). https://doi.org/10.1038/s41396-021-01088-5
o Kaul S;Choudhary M;Gupta S;Dhar MK; Engineering Host Microbiome for Crop Improvement and Sustainable Agriculture. Frontiers in microbiology, 12 (2021). doi: 10.3389/fmicb.2021.635917
o Ke J;Wang B;Yoshikuni Y; Microbiome Engineering: Synthetic Biology of Plant-Associated Microbiomes in Sustainable Agriculture. Trends in biotechnology, 39, 3 (2021). doi: 10.1016/j.tibtech.2020.07.008
- Please, indicate where Figures 4 (line 297) and 5 (line 541) are.
- Could the authors carefully revise the manuscript for grammatical inconsistencies and correct them to make sentences more straightforward? Consider breaking up those sentences that are split with several commas by rewriting them as two sentences (at times it is hard to keep track of the meaning as written). The English style needs to be reviewed to sound more scientific and professional.
- The Bibliography section was not homogeneously written. This made the reference search process difficult in some cases.
Minor comments:
· Line 4: Is an author missing at the end of the list?
· Lines 90-94, 195-199, 205-207, 219-221, 268-281: please consider moving this information to the Conclusion section, adequately referenced. I suggest the authors avoid subjective commentaries and reference every statement of the manuscript.
· Line 158: what is considered as “good or bad microbes”? References?
· Lines 164-174: references?
· Lines 217-219: please, rewrite this sentence. It is difficult to understand.
Author Response
Response to the Editor and reviewers
Dear Editor,
Thank you very much for editing our review article " Plant Microbiome Engineering: Hopes or Hypes " Manuscript-ID " biology-203419" and also thanks to the reviewers for giving us constructive suggestions which could significantly improve the quality of our paper. We have considered you’re and the reviewer's comments and modified our article according to your directions. The changes we made in the revised file are highlighted with track changes. The following paragraphs contain our responses to these comments.
Reviewer#1.
Comments and Suggestions for Authors
The review by Afridi et al. (biology-203419) “Plant Microbiome Engineering: Hopes or Hypes” makes an overview of plant microbiome composition, and describes the recent developments and achievements made around the challenges in plant microbiome engineering. The authors provide relevant information about this field here, however, this topic has been widely covered in science over the last few years. I, thereupon, wonder about the impact that it could have on the readers of /Biology/ since there are already many Reviews published about it.
I hope that my comments can help to get an improved version:
Major comments:
The abstract is too long and does not highlight the relevance and impact of this manuscript to be published. I encourage the authors to rewrite it and use shorter sentences for better understanding. Especially lines 30-45 appear repetitive and unclear.
o Lines 30-32: do not clarify the meaning of Rhizosphere microbiome
o Lines 32-34: please, avoid subjective statements.
Line 34: which argument?
Response. The abstract was re-write concisely and comprehensively and improved utterly. The aims and objectives explicitly explained as per the recommendation. All the above recommended sentences improved accordingly
- I found the structure of the Review quite unbalanced. Most of the manuscript is an Introduction, and the rest (2. Conventional approaches; 3. Conclusion and future remarks) is vaguely described and not well covered. What does this review offer that the others already published on this topic do not offer?
Response. The author(s) encourage the constructive comments on the review. The review composed and displayed very well. That’s true that a massive data has been published on the current topic. But every research and review article talks about its composition and the possible approach of plant microbiome engineering. The complexity, diversity and the challenges which could be countered during microbiome engineering utterly overlooked. This review article highlights the potential barriers of the biological, molecular and biotechnological approaches of the proposed microbiome engineering in already published literature, which might grab the attentions of the plant and molecular scientists and will be helpful in future
- I suggest the authors include the following published Reviews in this manuscript:
o Lau, SE., Teo, W.F.A., Teoh, E.Y. et al. Microbiome engineering and plant biostimulants for sustainable crop improvement and mitigation of biotic and abiotic stresses. Discov Food 2, 9 (2022). https://doi.org/10.1007/s44187-022-00009-5
o Albright, M.B.N., Louca, S., Winkler, D.E. et al. Solutions in microbiome engineering: prioritizing barriers to organism establishment. ISME J 16, 331–338 (2022). https://doi.org/10.1038/s41396-021-01088-5
o Kaul S;Choudhary M;Gupta S;Dhar MK; Engineering Host Microbiome for Crop Improvement and Sustainable Agriculture. Frontiers in microbiology, 12 (2021). doi: 10.3389/fmicb.2021.635917
- Ke J;Wang B;Yoshikuni Y; Microbiome Engineering: Synthetic Biology of Plant-Associated Microbiomes in Sustainable Agriculture. Trends in biotechnology, 39, 3 (2021). doi: 10.1016/j.tibtech.2020.07.008
Response. The references are added to the mentioned portion (Ref, 41,40, 69,70)
- Please, indicate where Figures 4 (line 297) and 5 (line 541) are.
Response. Corrected the mistakes in line# 316 and 571
- Could the authors carefully revise the manuscript for grammatical inconsistencies and correct them to make sentences more straightforward? Consider breaking up those sentences that are split with several commas by rewriting them as two sentences (at times it is hard to keep track of the meaning as written). The English style needs to be reviewed to sound more scientific and professional.
Response. The article revised thoroughly and improved its grammar
- The Bibliography section was not homogeneously written. This made the reference search process difficult in some cases.
Response. Corrected this section
Minor comments:
Line 4: Is an author missing at the end of the list?
Response. Corrected the author list
Lines 90-94, 195-199, 205-207, 219-221, 268-281: please consider moving this information to the Conclusion section, adequately referenced. I suggest the authors avoid subjective commentaries and reference every statement of the manuscript.
Response. The authors appreciated the suggestions and moved some information to the conclusion sections, such as 205-207, 219-221. As the conclusion utterly modified and improved and the recommended information now possess the conclusion but the rest of information
Line 158: what is considered as “good or bad microbes”? References?
Response. The good ones are beneficial microbes of the rhizosphere which are involved in plant growth promotion through nutrient uptake in plants, antagonism to plant pathogens, and plant tolerance against abiotic stresses. However, the bad ones are plant parasitic fungi and nematodes which cause diseases of economic importance in important crop plants and result in serious issues of reduction in productivity and food security. Reference.
Ali, M.A.; Naveed, M.; Mustafa, A.; Abbas, A. The Good, the Bad, and the Ugly of Rhizosphere Microbiome. In Probiotics and Plant Health; Kumar, V., Kumar, M., Sharma, S., Prasad, R., Eds.; Springer: Singapore, 2017; pp. 253–290 ISBN 978-981-10-3473-2.
- Lines 164-174: references?
Response. The references are added to the mentioned portion (Ref, 39,40)
- Lines 217-219: please, rewrite this sentence. It is difficult to understand.
Reviewer 2 Report
The authors explicitly described the complexity and diversity of plant microbiome, along with its hurdles and future perspectives. This review article I think openly discusses the challenges which face during the proposal of microbiome engineering. The author articulated and composed very well, though I have some recommendations which could improve the quality of the review article
· The aricile contains some typo errors, check throught.
· Line 35, interactions with?
· Line 34-37, the sentence is way long which makes it confusing, should be split for clarity.
· Line 40-42, How come conventional agricultural practices have been proposed as a new paradigm for harnessing plant microbiome engineering to disentangle agromanagement hurdles? In modern agriculture, conventional agricultural practices are being replaced with sustainable agriculture practices to minimize the effects of hazards on animals, plants, and microbiota, comes from conventional agricultural practices
· The language doesn’t sound scientific especially abstract and should be simplified so that the reader can understand without continuously looking for a dictionary. See the article for reference only https://www.mdpi.com/2079-7737/10/2/101
· Line 52, Keywords should be consistent and all small letters.
· Line 58-62. The sentence should be rephrased for clarity.
· Lines 83, 137, 152, 196, 220, 225, 253, 389, 397, 399, 409. should write the name of the author followed by the citation. Please go through the whole text.
· Line 208-210, rephrase it.
· Line 238, Should be consistent with the term plant “growth promoting bacteria” use unify term. Please go through the whole text.
· Line 243, Make sure it's a full stop or question mark.
· Line 243, 244. “To date, there is a lack of knowledge or research availability that show underground soil ecosystem” or “To date, there is a lack of knowledge or research gap to show underground soil ecosystem? Confirm it.
· Line 257, “carbon, C and N ratio” C?
· Line 280, Filed typo.
· Line 294, “PGPR” use full form and the abbreviated form in the parathesis when appearing first in the text and then solely use only abbreviated form onward.
· Line 316, figure citation in the text is inconsistent. Bold, normal
· Line 403, rephrase
· Line 658, Conclusion and future remarks. This portion could be improved and should discuss hurdles of plant microbiome engineering dominantly followed by eco-friendly strategy
Author Response
Response to the Editor and reviewers
Dear Editor,
Thank you very much for editing our review article " Plant Microbiome Engineering: Hopes or Hypes " Manuscript-ID " biology-203419" and also thanks to the reviewers for giving us constructive suggestions which could significantly improve the quality of our paper. We have considered you’re and the reviewer's comments and modified our article according to your directions. The changes we made in the revised file are highlighted with track changes. The following paragraphs contain our responses to these comments.
Reviewer#2.
The author Afridi et al. explicitly described the complexity and diversity of plant microbiome, along with its hurdles and future perspectives. This review article I think openly discusses the challenges which face during the proposal of microbiome engineering. The author articulated and composed very well, though I have some recommendations which could improve the quality of the review article
Minor Revision
The article contains some typo errors, check throughout
Response. # The article checked and corrected all the typo errors
Line 34, interactions with?
Response. The sentence corrected
Line 34-37, the sentence is way long which makes it confusing, should be split for clarity.
Response. The length of the sentences now reduced and improved it
Line 40-42, How come conventional agricultural practices have been proposed as a new paradigm for harnessing plant microbiome engineering to disentangle agromanagement hurdles? In modern agriculture, conventional agricultural practices are being replaced with sustainable agriculture practices to minimize the effects of hazards on animals, plants, and microbiota, comes from conventional agricultural practices
Response. The author raised a valid point, but here in microbial ecology, we are prioritizing the health and functionality of the belowground microbiome ecosystem. Our objective is to optimize the plant-associated beneficial microbe’s functions against plant pathogens and enhance microbial biodiversity. We urge that plant microbiome functions could be optimized and engineer via conventional agriculture practices, so the vitality of these practices couldn’t be overlooked, though, we entered into the advanced genomics era
Response.
The language doesn’t sound scientific especially abstract and should be simplified so that the reader can understand without continuously looking for a dictionary. See the article for reference only https://www.mdpi.com/2079-7737/10/2/101
Response. The abstract re-write and improved as per recommendation
Response. Line 52, Keywords should be consistent and all small letters.
Response. Keywords corrected
Line 58-62. The sentence should be rephrased for clarity.
Response. The sentence rephrased #89,90
Lines 83, 137, 152, 196, 220, 225, 253, 389, 397, 399, 409. should write the name of the author followed by the citation. Please go through the whole text.
Response. The missed text citation added and double checked throughout the article and rephrased
Line 208-210, rephrase it.
Response. Rephrased, line# 236-238
Line 238, Should be consistent with the term plant “growth promoting bacteria” use unify term. Please go through the whole text.
Response. Corrected through the whole text
Line 243, Make sure it's a full stop or question mark.
Response. It’s a question mark.
Line 243, 244. “To date, there is a lack of knowledge or research availability that show underground soil ecosystem” or “To date, there is a lack of knowledge or research gap to show underground soil ecosystem? Confirm it.
Response. Confirmed and rephrased line# 286-287
Line 257, “carbon, C and N ratio” C?
Response. Corrected and rephrased
Line 280, Filed typo.
Response. Corrected
Line 294, “PGPR” use full form and the abbreviated form in the parathesis when appearing first in the text and then solely use only abbreviated form onward.
Response. Corrected and followed
Line 316, figure citation in the text is inconsistent. Bold, normal
Response. The figures shows consistency now and maintained normal
Line 403, rephrase
Response. Rephrased
Line 658, Conclusion and future remarks. This portion could be improved and should discuss hurdles of plant microbiome engineering dominantly followed by eco-friendly strategy
Response. # The conclusion thoroughly modified and improved as per suggestion
Round 2
Reviewer 1 Report
Dear authors,
I was not able to find the new version of the manuscript with the corrections implemented. The file named "biology-2034195-peer-review-v2" belongs to a document other than the manuscript.
Author Response
We apologive for the inconvenience. The correct version of the manuscript has been attached now.

Round 3
Reviewer 1 Report
I thank the authors for the changes made in the manuscript and for its improvement in quality and rigor. I hope these comments can help to make it more accurate:
Major comments:
- I found unnecessary the addition of a Simple Summary. It mainly repeats what is explained in the abstract. The abstract itself should summarize enough of the content of the manuscript.
- Following the new structure in the manuscript, the section “3. Conclusion and future remarks” (line 706) should become point “9. Conclusion and future remarks”
Minor comments:
· Line 192: what is considered “good or bad microbes” for authors should be explained in the manuscript.
Author Response
Response to the Editor and reviewers
Dear Editor,
Thank you very much for editing our review article " Plant Microbiome Engineering: Hopes or Hypes " Manuscript-ID " biology-203419" and also thanks to the reviewers for giving us constructive suggestions which could significantly improve the quality of our paper. We have considered you’re and the reviewer's comments and modified our article according to your directions. The changes we made in the revised file are highlighted with track changes. The following paragraphs contain our responses to these comments.
Comments and Suggestions for Authors
I thank the authors for the changes made in the manuscript and for its improvement in quality and rigor. I hope these comments can help to make it more accurate:
Major comments:
- I found unnecessary the addition of a Simple Summary. It mainly repeats what is explained in the abstract. The abstract itself should summarize enough of the content of the manuscript.
Response. Thanks for the comment. The Simple Summary is the integral part of review article in biology journal. Yes, the summary, just portrays the concise and comprehensive form of abstract. This is why, the simple summary is the summarize form of the whole concept and abstract.
- Following the new structure in the manuscript, the section “3. Conclusion and future remarks” (line 706) should become point “9. Conclusion and future remarks”
Response. section “3. Conclusion and future remarks” (line 706) replaced with “9. Line# 645
Minor comments:
- Line 192: what is considered “good or bad microbes” for authors should be explained in the manuscript.
Response. The good or bad microbes elaborated as per suggestion (line# 165-171)